# Spontaneous Flora as Reservoir for the Survival and Spread of the Almond Anthracnose Pathogen (*Colletotrichum godetiae*) in Intensive Almond Orchards

**DOI:** 10.3390/plants14121762

**Published:** 2025-06-09

**Authors:** Madalena Ramos, Rodrigo Maurício, Vicelina Sousa, Pedro Talhinhas

**Affiliations:** 1Linking Landscape, Environment, Agriculture and Food Research Centre (LEAF), Associate Laboratory TERRA, Instituto Superior de Agronomia, Universidade de Lisboa, Tapada da Ajuda, 1349-017 Lisboa, Portugal; madalenaramos@isa.ulisboa.pt; 2Instituto Superior de Agronomia, Universidade de Lisboa, Tapada da Ajuda, 1349-017 Lisboa, Portugal; rodrigomauricio271507@gmail.com; 3Forest Research Centre (CEF), Associate Laboratory TERRA, Instituto Superior de Agronomia, Universidade de Lisboa, Tapada da Ajuda, 1349-017 Lisboa, Portugal; vsousa@isa.ulisboa.pt

**Keywords:** inoculum reservoirs, splash dispersion, pathogenicity, secondary conidia, vegetation management

## Abstract

Almond anthracnose, primarily caused by *Colletotrichum godetiae*, severely affects intensively irrigated almond orchards. This polyphagous pathogen is dispersed among plants by rain splashes. Consequently, weeds may contribute to the survival and dispersal of the inoculum during the almond tree’s dormant period. This study investigated how *C. godetiae* interacts with plants from various species in the spontaneous flora of almond orchards and how these plant species may influence the maintenance and spread of inoculum and the disease. After inoculating a collection of plants with *C. godetiae* conidia, it was observed that the fungus can cause symptoms and signs on *Lathyrus tingitanus* and on *Trifolium pratense* and act as an epiphyte with the ability to maintain and multiply conidia on *Conyza canadensis*, *Medicago orbicularis*, *Polygonum aviculare*, *Scorpiurus sulcatus*, *Taraxacum officinale*, and *Trifolium vesiculosum*, thus contributing to the survival and multiplication of the inoculum. Conidia germinated and produced appressoria on *Andryala integrifolia*, *Cichorium intybus*, *Medicago polymorpha*, *Medicago sativa*, *Torilis arvensis*, *Picris echioides,* and *Rumex pulcher,* but no further development was detected, suggesting that these plants may limit the spread of the pathogen. A better understanding of the susceptibility of almond orchard flora will support optimized vegetation management to reduce inoculum reservoirs.

## 1. Introduction

Almond trees (*Prunus dulcis* (Mill.) D.A. Webb) are cultivated on most continents, attaining great economic and socio-cultural significance in the Mediterranean basin, where this stone fruit crop is traditionally grown [1,2,3]. Almond anthracnose, caused by fungi of the genus *Colletotrichum*, is one of the diseases that severely affect the productivity of this crop [4]. The most characteristic symptoms of almond anthracnose are seen as early as on unripe fruit, with the formation of depressed, round, orange lesions and the production of gummosis. Whenever the symptoms progress, whitish mycelium and masses of orange conidia appear on the surface of the infected kernels [2]. The species *Colletotrichum godetiae* Neerg, *C. acutatum* J.H. Simmonds, *C. fioriniae* (Marcelino & Gouli) Pennycook, *C. nymphaeae* Pass. and *C. simmondsii* R.G. Shivas & Y.P. Tan (members of the acutatum species complex) have already been reported as pathogens of almond anthracnose [5,6,7,8]. However, *C. godetiae* and *C. acutatum* are reported as the prevalent species causing anthracnose in the main almond production regions of the world, such as California (USA) [4,9], South Australia [10,11], Andalusia (Spain) [6], and Alentejo (Portugal). In Alentejo *C. godetiae* is the predominant causal agent [12].

The pathogen mainly affects the fruit, but it is also capable of infecting flowers, leaves, and woody branches. Fruits affected by the disease at the end of the almond tree’s phenological cycle that remain in the canopy, known as mummified almonds, are capable of releasing conidia at constant rates for six months [13,14]. Mummified almonds are major reservoirs of inoculum during the dormant period, contributing to the onset of primary infections that begin during late winter and early spring when the rains occur in early phenological stages of the crop. In *Colletotrichum*, conidia are the fungus fungus’s main form of dispersal. High humidity conditions, with frequent rainfall and warm temperatures, favour the maintenance and dispersal of conidia, as this occurs through rain splashes [15,16,17]. Although there are differences in the way conidia act on fruit, leaves, and flowers, they are capable of germinating and forming germ tubes. In some organs, such as leaves, these tubes can be long and lead to the formation of secondary conidia [16]. After the germination of the conidia, the formation of the appressorium can occur, a structure that allows the fungus to penetrate the surface of the organ and begin its endophytic activity [2,18]. The colonization strategy of the anthracnose pathogen on almond tissues is described as subcuticular-intracellular hemibiotrophic and intercellular necrotrophic [19]. This strategy, as outlined by the authors, is characterized by the combination of a biotrophic phase that occurs after penetration of the cuticle and includes the production of infection vesicles and primary hyphae that spread through the cells of the epidermis and mesophyll, with the host cells remaining alive. After this period, the formation of secondary inter- and intracellular hyphae that secrete specific enzymes for the degradation of cell walls leads to the appearance of visible symptoms, such as necrotic areas that characterize the necrotrophic phase, where the host cells die.

Most species of *Colletotrichum*, and specifically those associated with almond anthracnose, are polyphagous and therefore have little or no pathogenic specialization, which is why they can infect other hosts, such as other species of stone and pome fruit, vegetables, cereals, citrus fruit, olive trees, among others, which can contribute to cross-infections [6] and thus increase the mobility of the pathogen between agricultural crops [20,21]. In this case, rain splash is a disease dispersal process, whose efficiency strongly increases with high density of host plants [22,23]. Considering the phenological cycle of the almond tree, the fruit remains on the plant from February/March to June/July, and the leaves fall during Autumn, budding again in Spring [24], meaning that the target organs of the almond anthracnose pathogen are not always available in the orchard. However, the ability to colonize different hosts gives the pathogen the chance to survive on alternative hosts that are part of the flora surrounding the crops usually affected.

The likelihood of weeds appearing in almond orchards is high, where herbaceous vegetation develops mainly during the winter and late September or early October, coinciding with the increase in rainfall [25]. Spontaneous vegetation can be divided into annual, biennial, and perennial plants. Annual plants are divided into summer annuals and winter annuals; the former germinate at the end of winter and go through their biological cycle in summer, with seeds ripening at the end of summer. The success of these plants depends on the availability of water, making them important weeds in irrigated agriculture. Winter annuals germinate after the first autumn rains and develop vegetatively in winter, appearing during the almond orchard’s dormant period. Biennial plants reproduce only by seeds and experience growth arrest when exposed to water stress or winter cold. Perennial plants reproduce by seeds and vegetatively (including meristems, bulbs, tubers, and rhizomes) [26].

Due to the disadvantages of bare soil between orchard rows, such as increased erosion, compaction, and reduced water infiltration into the soil [25], many agricultural engineers choose to maintain vegetation cover in the almond orchards. Natural vegetation cover uses the spontaneous flora that occurs in the orchard to provide diverse ecosystem services, including pollination, water and soil retention, and traction for machinery. For example, Poaceae plants have been shown to control erosion and improve the accessibility of agricultural equipment in irrigated almond orchard with clay soils [27]; Brassicaceae plants are used as green manure to improve the organic matter content in the soil [28]; Fabaceae plants perform nitrogen fixation thus improve soil fertility [29,30].

The interaction between spontaneous flora and orchards also affects the dynamics of interactions between flora and pathogens. The presence of a variety of plant species in the orchard ecosystem may act as a reservoir for fungal pathogens and may have a potential influence on the success of infection and progression of disease. It is noteworthy that species of *Colletotrichum*, such as *C. godetiae*, are known to infect a wide variety of host plants. For example, conidia of *Colletotrichum* species responsible for strawberry anthracnose have been identified asymptomatically on weeds *Conyza* sp. and *Vicia* sp. [31,32]. Therefore, understanding the susceptibility of these species to *C. godetiae* is essential to assess their possible role in the persistence and spread of the pathogen in almond orchards.

This study aims to understand how *C. godetiae* interacts with the different herbaceous plants present in almond orchards, in order to show which weeds have the capacity to act as inoculum reservoirs. The study is conducted on intensive (250–600 trees/ha) and super-intensive (>600 trees/ha) almond orchards cultivated under Mediterranean climate at the Alentejo region of Portugal, an area that has experienced a high increase in almond cultivation in recent years totalling 31,526 ha in 2023 [33] accompanied by an increase in anthracnose incidence and severity [12].

## 2. Results

### 2.1. Identification of Plant Species Present in Almond Orchards in Alentejo

The list of plant species reported at locations where almond orchards are currently installed (Figure 1) revealed a high species diversity. A total of 405 different herbaceous species were compiled (Table 1; Appendix A). The predominant botanical families are Asteraceae, Fabaceae, Poaceae, Apiaceae, and Caryophyllaceae, which comprise 74, 55, 47, 21, and 19 species. The most common genera are Trifolium, Medicago, Silene, and Plantago.

The 10 species with the greatest distribution in almond orchards in Alentejo are, in decreasing order, *Calendula arvensis* L., *Crepis vesicaria* L., *Foeniculum vulgare* Mill., *Echium plantagineum* L., *Briza maxima* L., *Cistus salviifolius* L., *Stachys arvensis* L., *Galactites tomentosus* Moench, *Sonchus oleraceus* L., and *Anagallis arvensis* L. sensu Franco & Rocha Afonso. The most frequent weeds in the almond orchards of Beja, Évora, Portalegre, and Setúbal are *Calendula arvensis*, *Geranium molle* L., *Foeniculum vulgare,* and *Cistus salviifolius*, respectively.

### 2.2. Development of the Colletotrichum godetiae Infection Process and Symptoms on Inoculated Leaves

One day after inoculation and upon counting 100 conidia for each polish replica, conidia were found to have germinated and developed appressoria in all weed species (Figure 2). Concerning the formation of appressoria, only *Polygonum aviculare*, *Taraxacum officinale*, Trifolium pratense, and *Scorpiurus sulcatus* did not have conidia with appressoria formation. The highest rate of appressoria formation was recorded on *Andryala integrifolia* (34%). The highest spore germination rate was found on *Polygonum aviculare* (49%), while the lowest germination rate was recorded on *Trifolium vesiculosum* (9%). Seven days after inoculation, fungal conidia germinated and appressoria formed on all plants.

When comparing the numbers of germinated conidia and formed appressoria obtained for the first and seventh day after inoculation, there was an evolution in the rate of germination and appressoria formation (Figure 2 and Figure 3). Bearing in mind that for some species it was not possible to count 100 conidia in the SEM images, it was necessary to project the values obtained to 100 conidia. Concerning germination rates, all species, except *Picris echioides*, recorded higher numbers of germinated conidia. *Scorpiurus sulcatus*, *Medicago sativa* and *Taraxacum officinale* stood out for the greater difference in conidia observed between the two periods after inoculation (81, 68, and 72 more germinated conidia, respectively, compared to the first day). For most of the inoculated species, higher values of conidia with appressorium formation were observed, except for *Cichorium intybus*, *Picris echioides,* and *Trifolium vesiculosum*, which showed lower values compared to the first day, while *Medicago polymorpha* and *Medicago sativa* showed the same value.

Considering the size of the germ tubes and the number of germ tubes that developed per conidium, it was found that the germ tubes of *Colletotrichum godetiae* were longer on *Trifolium vesiculosum* and *Picris echioides* compared to the other species, and in both cases, the conidia only produced one germ tube (Table 2). The conidia that produced the shortest germ tubes were *Andryala integrifolia*, *Lathyrus tingitanus,* and *Torilis arvensis*. However, the average length was not significantly different from the other plant species, except for *Trifolium vesiculosum*. In these three plants, the rate of appressoria formation in relation to the number of germinated conidia was over 75%, suggesting that the fungus was not constrained in forming the first structures of infection. The highest number of conidia that produced more than one germ tube was recorded on *Medicago orbicularis*, with a significantly higher average value compared to the other hosts.

Conidia with sessile appressorium were found on *Lathyrus tingitanus*, *Torilis arvensis*, *Andryala integrifolia* (Figure 3B), Conyza canadensis, *Rumex pulcher*, *Polygonum aviculare* and *Scorpiurus sulcatus*. It should be noted that *Lathyrus tingitanus* showed a high rate of appressoria formation (81%) (Figure 3A), approximately half of which were sessile.

Throughout the collection of plants studied, *Colletotrichum godetiae* conidia produced more than one germ tube on seven plants (Table 2). The highest number of conidia with two germ tubes was found on *Medicago orbicularis*, while on *Andryala integrifolia*, *Cichorium intybus*, Conyza canadensis, *Lathyrus tingitanus*, *Picris echioides*, *Polygonum aviculare*, *Rumex pulcher,* and *Trifolium vesiculosum,* the fungal spores generated a single germ tube exclusively, indicating a possible variation in germination between the different host species.

The formation of secondary conidia was also observed 7 days after inoculation (Figure 4). Secondary conidiogenesis was observed on seven plants, namely: Conyza canadensis, *Lathyrus tingitanus* (Figure 4D), *Medicago orbicularis* (Figure 4C), *Polygonum aviculare* (Figure 4B), *Scorpiurus sulcatus* (Figure 4A), *Taraxacum officinale,* and *Trifolium vesiculosum*. *Medicago orbicularis* was the host with the highest percentage of secondary conidia among the total number of germ tubes counted in that species.

After inoculation, the plants were monitored to assess the appearance of possible anthracnose symptoms. For Conyza canadensis, *Cichorium intybus*, M. sativa, *Picris echioides*, *Scorpiurus sulcatus*, *Torilis arvensis*, Trifolium pratense, and *Trifolium vesiculosum*, although germination and the formation of conidial appressorium were observed, they did not develop symptoms or masses of conidia on the surface when subjected to a humid chamber, nor did fungal colonies of *Colletotrichum godetiae* develop in the PDA from which leaves were isolated with and without disinfection.

*Andryala integrifolia*, *Cichorium intybus*, *Medicago polymorpha*, *Medicago sativa*, *Picris echioides*, *Rumex pulcher* and *Torilis arvensis* despite showing significant rates of germination and appressoria formation, did not develop symptoms or confirm the presence of the fungus throughout the trial and symptom development was also not observed when the leaves of each species were kept in a humid chamber and placed on PDA.

*Lathyrus tingitanus*, *Polygonum aviculare*, and *Taraxacum officinale* were the only species that showed chlorotic spots on the margins of the leaves 7 days after inoculation and, when placed in a humid chamber, orange masses of conidia developed on the surface of the leaves, which confirmed the presence of *Colletotrichum godetiae* when reisolated onto PDA and the conidia were observed under an optical microscope. In addition, fungal colonies with a light grey colour and orange conidial masses developed on the plates where the leaves were isolated with and without 1% NaClO disinfection. *Taraxacum officinale* did not develop symptoms or show conidia on the leaf surface when placed in a humid chamber, and in the Petri dish where *Taraxacum officinale* leaves were isolated with disinfection, the presence of the fungus was confirmed after isolating onto PDA, with the development of light grey fungal colonies. After inoculation, no symptoms were observed on the leaves of *Polygonum aviculare*, nor were there any symptoms or masses of conidia when the leaves were kept in a humid chamber. However, when leaves without disinfection were isolated on PDA, the presence of *Colletotrichum godetiae* was confirmed, with the development of white fungal colonies with orange spore masses. Trifolium pratense showed no symptoms of anthracnose throughout the trial, but when the leaves were placed in a humid chamber, masses of orange spores developed on the plant tissues. For *Lathyrus tingitanus*, *Polygonum aviculare*, *Taraxacum officinale,* and Trifolium pratense, the morphological characteristics of the conidia were confirmed under a compound optical microscope. The conidia were fusiform and hyaline.

## 3. Discussion

The plants selected for this work are a sample of the spontaneous flora, which are often found in almond orchards or the surrounding agricultural fields. Although some of the species that make up this flora are not economically useful, some genera, such as *Medicago* or *Trifolium*, are important nitrogen-fixing species in the soil and can occur between the orchard rows. Between the rows of vineyards and orchards, intercropping is a common practice that can be natural or sown. It helps to improve aeration, infiltration, and water retention in the soil, improves organic matter content, controls erosion, attracts auxiliary arthropods, and promotes carbon sequestration and nitrogen fixation in the soil [35]. In almond orchards, most farmers choose to keep the weeds between the rows, except at the time of the almond harvest [25], which means that the almond trees are in contact with other plant species for most of the phenological cycle. The continuous interaction between these plant species and the orchard environment may influence not only soil health and biodiversity but also the dynamics of plant–pathogen interactions, shaping the susceptibility of different species to fungal infection.

On the first day after inoculation, for all plant species, the fungus germinated, and only *Polygonum aviculare* and *Scorpiurus sulcatus* did not have conidia forming an appressorium, which may indicate the high efficacy of *Colletotrichum godetiae* in the first phase of infection for the inoculated species. Seven days after inoculation, there was a significant increase in the number of germinated and appressoria-forming conidia in most species, except *Cichorium intybus*, *Picris echioides,* and *Trifolium vesiculosum*, which may indicate that for these species, the survival of the inoculum is reduced. Of all the species inoculated, only four confirmed the presence and development of *C. godetiae* (*Lathyrus tingitanus*, *Polygonum aviculare*, *Taraxacum officinale,* and *Trifolium pratense*). During the experiment, *Lathyrus tingitanus* showed symptoms of infection with the pathogen, with chlorotic spots appearing on the leaves. However, the formation of conidia masses on the surface of leaves and stems was enhanced by placing the plant in a humid chamber.

Only germinated conidia with a single germ tube were found on eight plants. However, this parameter did not appear to be related to any of the other criteria used in this study for evaluating the ability of the pathogen to infect the plant.

When leaves of *Lathyrus tingitanus*, *Polygonum aviculare*, and *Taraxacum officinale* were subjected to isolation on PDA, with and without disinfection, the pathogen only developed on *Lathyrus tingitanus*. This suggests that the fungus acted as an epiphyte, but also that the process of infection had already begun [16,36]. In *Polygonum aviculare* and *Trifolium pratense,* the fungus was confirmed on leaves without disinfection only, which indicates the fungus’s epiphytic behaviour, without successfully developing the infection process. For *Taraxacum officinale*, the development of *Colletotrichum godetiae* colonies on PDA was confirmed only on disinfected leaves, which proves that the fungus had already started the infection process, even though no symptoms or superficial masses of conidia were seen. This case may suggest that the fungus may have opted for a latency period, defined as a prolonged period in the pathogen’s life cycle, during which it lies dormant inside the host until it switches to an active phase, forming lesions on the plant [37,38,39].

*Andryala integrifolia*, *Lathyrus tingitanus*, and *Torilis arvensis* were the species where the fungus produced the shortest germ tubes and the greatest number of sessile appressoria, which suggests that for these hosts, the conidia did not need to look for new sites to form an appressorium and infect the plant. These conidia ceased germ tube growth and formed appressoria.

Although few species showed symptoms and signs after inoculation by *Colletotrichum godetiae*, the fungus was able to produce secondary conidia in seven of the 15 species under study. Because of this, *C. godetiae*, as an alternative to the inability to infect the host, chooses to increase the reservoirs of inoculum to increase the possibility of dispersal and thus colonize other species that are more favourable to its survival [36,40]. However, *Andryala integrifolia*, *Cichorium intybus*, *Medicago polymorpha*, *Medicago sativa*, *Picris echioides*, *Rumex pulcher*, *Torilis arvensis,* and *Trifolium pratense* did not produce secondary conidiogenesis. The failure to produce secondary conidia on *Trifolium pratense* can be explained by the fact that it is susceptible to *C. godetiae*. However, *Andryala integrifolia*, *Cichorium intybus*, *Medicago polymorpha*, *Medicago sativa*, *Picris echioides*, *Rumex pulcher*, and *Torilis arvensis* did not produce symptoms or signs and did not allow the fungus to develop, which suggests that they may have a suppressive action on the spread of the pathogen’s inoculum, making them plants with a low level of susceptibility to the disease. *Lathyrus tingitanus*, *Medicago orbicularis*, *Polygonum aviculare,* and *Scorpiurus sulcatus* are spring and summer annuals [26,30], found in almond orchards during fruit development, allowing the inoculum to be kept in the field during the almond tree’s most susceptible phenological phase [17,41]. *Cichorium intybus*, *Conyza canadensis*, *Trifolium pratense*, and *Taraxacum officinale* are perennial species, and *Medicago polymorpha* and *Trifolium vesiculosum* [26,30] are annual herbaceous species, but these six species are found in the field between fall and winter, making it possible to maintain the inoculum of *C. godetiae* during the dormant phase of the almond tree, due to the pathogen’s ability to produce secondary conidia on these hosts.

## 4. Materials and Methods

### 4.1. Identification of Plant Species Present in Almond Orchards in Alentejo

To list the weed species that occur in the distribution and spatial context of the almond orchards in Alentejo, the Flora-on database was used, which is a portal containing systematized information (photographic, geographical, morphological, and ecological) on the native and invasive species listed for the flora of Portugal https://flora-on.pt/ (accessed on 3 November 2024) [42]. The distribution of plant species in Flora-on is mapped using a 10 km UTM (Universal Transverse Mercator) square grid. The geographical information with the spatial distribution of the plant species was delimited using the shapefile of the almond parcels registered in 2023 with the Instituto de Financiamento da Agricultura e Pescas (IFAP) [34], which is available at https://www.ifap.pt/isip/ows/ (accessed on 3 November 2024). For this purpose, the intersection geoprocessing tool in the QGIS v.3.16.12 software was used to obtain the list of herbaceous species already registered in the geographical area where the almond trees are currently installed. Thereafter, perennial woody plant species and those reported only once in the region covered by the shapefile were excluded from the resulting file. Only the herbaceous species included in the Flora-on the square grid which overlaps the almond orchard areas, were obtained. In this way, the percentage of the distribution of the flora in relation to the total number of almond orchards in the districts of Beja, Évora, Portalegre, and Setúbal could be determined based on the presence or absence of each species in the network.

### 4.2. Cultivation of Selected Plants

For this study, 15 weed species present in Alentejo almond orchards were selected on the basis of the list drawn up in this paper (Table 1): *Andryala integrifolia* L., *Cichorium intybus* L., *Conyza canadensis* (L.) Cronquist, *Lathyrus tingitanus* L., *Medicago orbicularis* (L.) Bartal., *Medicago polymorpha* L., *Medicago sativa* L., *Picris echioides* L., *Polygonum aviculare* L., *Rumex pulcher* L., *Scorpiurus sulcatus* L., *Taraxacum officinale* (Weber), *Torilis arvensis* (Huds.) Link, *Trifolium pratense* L. and *Trifolium vesiculosum* Savi. The seeds used were harvested between 2017 and 2023, dried, and frozen for 24 h at −20 °C as a form of disinfection. To overcome the dormancy of leguminous seeds due to their tough tegument, the seeds were scarified by cutting through the tegument with a blade. For each species, 20 seeds were sown in sandy loam soil. The plants were kept in a greenhouse, in a controlled environment, at a temperature of 23 °C and under conditions of 12 h/day of light.

### 4.3. Inoculation of Plant Material with Colletotrichum godetiae

The *C. godetiae* isolate MR012 was grown in Potato Dextrose Agar (PDA, BD-Difco, Sparks, MD, USA) at 25 °C under darkness for 10 days. The conidia suspensions were prepared immediately before inoculating the plants. Using a scraper, the fungal colonies that grew on the PDA were scraped off to facilitate the release of spores into distilled, sterilized water. After this, the suspension obtained was filtered through a No.1 glass filter. The concentration of the conidia suspension was adjusted to 1 × 10^6^ conidia/mL using a Neubauer chamber.

Two inoculation methods were used to assess the progression and development of the pathogen. The plants were sprayed with approximately 15 mL of conidial suspension using a glass sprayer to evaluate the development of possible plant symptoms throughout the experiment. For microscopic monitoring of the fungus, two to three drops of 15 µL of conidial suspension were placed on random leaves of each species with a micropipette in areas previously indicated by a marker. In the first 24 h after inoculation, the inoculated plants were kept in humid chambers inside the greenhouse at a temperature of 23 °C. Experimental replicates were prepared by analyzing at least five different leaves per plant.

### 4.4. Germination of Conidia and Fungal Development on Inoculated Leaves

The conidia germination and formation of appressoria were observed using the nail polish method [43] and through images captured by scanning electron microscopy. Nail polish replicas were made one day after inoculation (dai). After the polish dried, the replicas were stained with cotton blue for 15 min, and preparations were made for observation under a compound optical microscope.

Leaves inoculated by spraying were observed under a scanning electron microscope (SEM) after 7 days at different magnifications. To do this, leaf fragments measuring around 25 mm^2^ were fixed with the adaxial side facing upwards on an aluminum support using double-sided carbon adhesive tape and then observed in the TM3030Plus benchtop SEM (Hitachi, Tokyo, Japan), equipped with image acquisition equipment at a voltage of 15 kV at different magnifications. From the images obtained, the germination rate, the rate of appressoria formation, and the rate of secondary conidiogenesis were determined, as well as quantifying the length of the germ tube using ImageJ v.2 software (National Institutes of Health, Boston, MA, USA).

Quantitative data for calculating the germination rate and the rate of appressoria formation were obtained by counting 100 conidia. For every 100 conidia, the conidia that germinated were counted, and those that formed appressoria were counted. The rates of appressoria formation were calculated as the ratio between the number of conidia with appressoria and the number of conidia that germinated. Secondary conidiogenesis rates were determined by the ratio between the number of germ tubes germinated and the number of secondary conidia formed, considering a set of 30 conidia.

### 4.5. Symptomatic Development and Confirmation of the Presence of Colletotrichum godetiae

After inoculation by spraying, the plants were monitored weekly for one month to detect the appearance of symptoms. Leaves with and without symptoms were detached from the plants and kept in a humid chamber at room temperature. In addition, leaves with and without symptoms were selected for fungal isolation on culture medium (PDA + Chloramphenicol 250 μg/mL). Leaves without symptoms were subjected to isolation without surface disinfection to account for epiphytic colonization. The leaves with symptoms were washed and disinfected using the protocol described by López-Moral et al. [6]. The surface of the leaves was disinfected by immersion in 1% sodium hypochlorite (NaClO) for 1 min, then washed in distilled water for 30 s and dried on absorbent paper. After isolation, the plates were incubated at 25 °C in the dark and kept for 15 days after incubation. The presence of *C. godetiae* on the surface of the plants in the humid chamber and in the isolations in PDA was assessed by observing the fungal colonies using a Leica MZ 12 binocular magnifying glass (Leica, Wetzlar, Germany) after the fungal colonies had grown.

In addition, observations were also made under the Leica DM 2500 compound optical microscope (Leica, Wetzlar, Germany) to confirm *Colletotrichum godetiae* through the morphological characteristics of its conidia.

### 4.6. Statistical Analysis

Data on germ tube length and the number of germinated germ tubes per conidium obtained from the SEM images were treated statistically using analysis of variance (ANOVA). The Tukey-HSD test (*p* < 0.05) was applied to test for significant differences between the means of the variables. Statistical analysis was carried out using RStudio v.4.1.2 software (Posit PBC, Boston, MA, USA).

## 5. Conclusions

The present study shows that *Colletotrichum godetiae* can interact with various species of spontaneous flora present in Alentejo almond orchards and shows that these species may play different roles in the epidemiology of almond anthracnose.

Species susceptible to infection were identified, namely *Lathyrus tingitanus*, *Polygonum aviculare*, *Taraxacum officinale,* and Trifolium pratense, in which symptoms and signs were observed. The species Conyza canadensis, *Medicago orbicularis*, *Scorpiurus sulcatus*, *Polygonum aviculare,* and *Trifolium vesiculosum* did not show symptoms of disease, but developed fungal structures on the leaf surface, namely the production of secondary conidia, suggesting their potential role as epiphytic reservoirs of inoculum.

On the other hand, plants such as *Andryala integrifolia*, *Cichorium intybus*, *Medicago polymorpha*, *Medicago sativa*, *Picris echioides*, *Rumex pulcher* and *Torilis arvensis* have been shown to allow the germination of conidia and the formation of appressoria, but without an infectious process, which suggests that they act as a barrier to proliferation of the pathogen in the orchard.

The differences observed between species in conidium germination, formation of appressoria, production of secondary conidia, and expression of symptoms reflect different levels of compatibility and susceptibility, reinforcing the diversity of plant–pathogen interactions that can be found in the almond orchard agro-ecosystem. These results underline the importance of spontaneous vegetation as a critical factor for survival, multiplication, and dispersal of the inoculum during periods when the susceptible organs of the almond tree are not available. Including this information in orchard vegetation management strategies can help mitigate inoculum pressure and reduce disease incidence, since some weed species may be regarded as suppressive towards the survival of inoculum.

Future research should further improve our understanding of the viability and persistence of secondary conidia on different species of spontaneous flora and extend the study to other species from botanical families that predominate in Alentejo’s almond orchards, such as Asteraceae, Fabaceae, Poaceae, Apiaceae and Caryophyllaceae, as well as other plants commonly sown between the rows of almond trees to improve their rooting, such as *Avena strigosa*, *Brassica napus*, *Ornithopus sativa*, *Sinapsis alba* and *Trifolium incarnatum*. The application of molecular and transcriptomic approaches will also provide insight into the metabolic mechanisms involved in host plant resistance or permissiveness, contributing to the development of more effective and sustainable integrated disease management practices.

## Figures and Tables

**Figure 1 plants-14-01762-f001:**
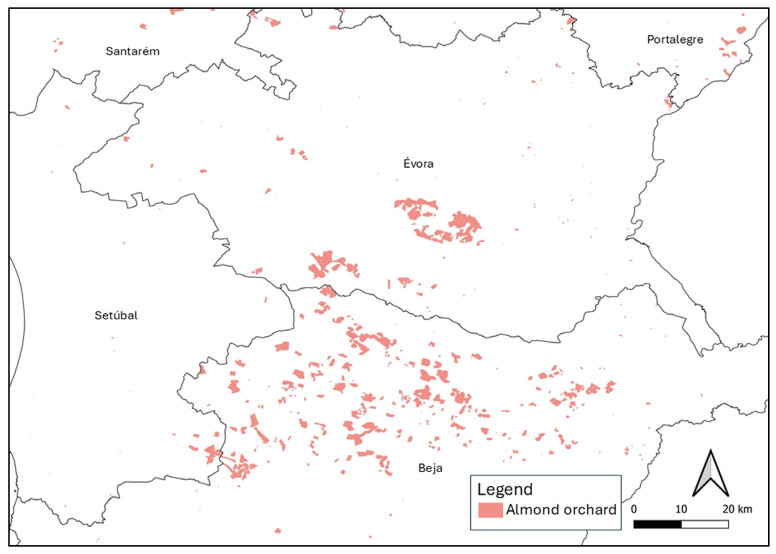
Distribution of the area of almond trees registered in 2023 with the Instituto de Financiamento da Agricultura e Pescas in the Alentejo region (Portugal) [34].

**Figure 2 plants-14-01762-f002:**
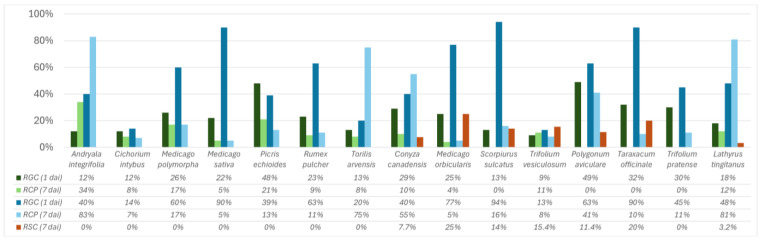
Rate of germinated conidia (RGC) and conidia with appressoria (RCP) one day after inoculation (1 dai) by observing nail polish replicas under a compound optical microscope. Rate of germinated conidia, rate of conidia with appressoria, and rate of secondary conidiogenesis (RSC) were recorded seven days after inoculation (7 dai) using images obtained by scanning electron microscopy.

**Figure 3 plants-14-01762-f003:**
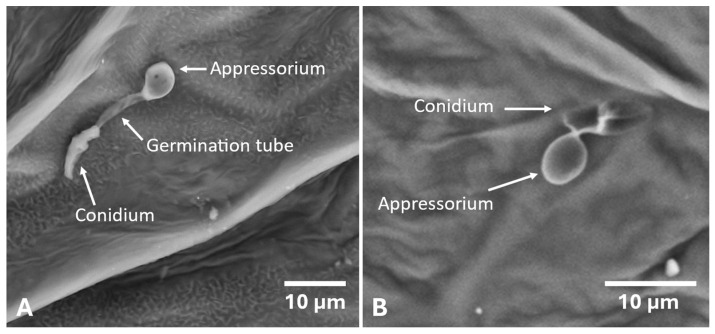
Germination of conidia and formation of *Colletotrichum godetiae* appressorium on the leaf surface of *Lathyrus tingitanus* (**A**). Sessile appressoria of *Colletotrichum godetiae* on the leaf surface of *Andryala integrifolia* (**B**). Fungal structures were observed by scanning electron microscopy seven days after inoculation.

**Figure 4 plants-14-01762-f004:**
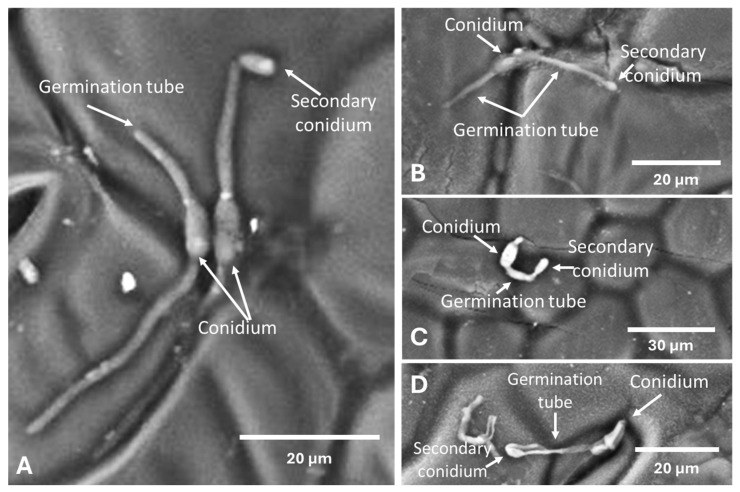
Production of secondary conidia of *Colletotrichum godetiae* on the leaf surface of *Scorpiurus sulcatus* (**A**), *Polygonum aviculare* (**B**), *Medicago orbicularis* (**C**), and *Lathyrus tingitanus* (**D**). Conidia with two germ tubes on *Scorpiurus sulcatus* (**A**). Fungal structures were observed by scanning electron microscopy seven days after inoculation.

**Table 1 plants-14-01762-t001:** List of herbaceous plant species used in this study and the respective percentage of their presence in the geographical area covered by the almond orchards in the districts of Beja, Évora, Portalegre, and Setúbal, which comprise the Alentejo region.

Botanical Family	Species	Beja (%)	Évora (%)	Portalegre (%)	Setúbal (%)	Total Number of Reports (%)
Apiaceae	*Torilis arvensis*	18.3	23.9	58.3	17.6	24.7
Asteraceae	*Andryala integrifolia*	33.8	43.5	37.5	35.3	37.3
	*Cichorium intybus*	22.5	21.7	45.8	41.2	27.8
	*Conyza canadensis*	2.8	2.2	4.2	11.8	3.8
	*Picris echioides*	23.9	8.7	12.5	17.6	16.5
	*Taraxacum officinale*	30.3	0.0	0.0	27.8	13.5
Fabaceae	*Lathyrus tingitanus*	4.2	34.8	8.3	11.8	14.6
	*Medicago orbicularis*	31.0	2.2	0.0	11.8	15.2
	*Medicago polymorpha*	43.7	39.1	45.8	29.4	38.6
	*Medicago sativa*	0.0	2.2	0.0	5.9	1.3
	*Scorpiurus sulcatus*	39.4	10.9	12.5	17.6	24.1
	*Trifolium pratense*	1.4	2.2	16.7	11.8	4.4
	*Trifolium vesiculosum*	1.4	6.5	0.0	5.9	3.2
Polygonaceae	*Polygonum aviculare*	5.6	2.2	20.8	17.6	8.2
	*Rumex pulcher*	19.7	13.0	33.3	17.6	19.0

**Table 2 plants-14-01762-t002:** Rate of sessile appressoria, average size of germ tubes, and average number of germ tubes per conidium recorded seven days after inoculation using images obtained by scanning electron microscopy.

Species	7 Dai		
	Average Germ Tube Size (µm)	Average Number of Germ Tubes/Conidium	Rate of Sessile Appressoria
*Torilis arvensis*	218.3 ± 0.4 a *	1.1 ± 0.3 ab *	31.3%
*Andryala integrifolia*	183.0 ± 0.2 a	1.0 ± 0.0 a	20.0%
*Cichorium intybus*	314.4 ± 0.2 a	1.0 ± 0.0 a	0%
*Conyza canadensis*	327.7 ± 0.4 a	1.0 ± 0.0 a	19.2%
*Picris echioides*	658.1 ± 0.2 ab	1.0 ± 0.0 a	0%
*Taraxacum officinale*	438.0 ± 0.3 a	1.4 ± 0.5 ab	0%
*Lathyrus tingitanus*	208.3 ± 0.4 a	1.0 ± 0.0 a	48.4%
*Medicago orbicularis*	439.8 ± 0.4 a	1.7 ± 0.5 c	0%
*Medicago polymorpha*	519.3 ± 0.4 a	1.1 ± 0.4 ab	0%
*Medicago sativa*	348.5 ± 0.3 a	1.3 ± 0.5 ab	0%
*Scorpiurus sulcatus*	237.9 ± 0.2 a	1.4 ± 0.5 b	4.7%
*Trifolium pratense*	373.4 ± 0.3 a	1.2 ± 0.4 ab	0%
*Trifolium vesiculosum*	1049.7 ± 0.8 b	1.0 ± 0.0 a	0%
*Polygonum aviculare*	354.1 ± 0.3 a	1.0 ± 0.0 a	8.6%
*Rumex pulcher*	530.3 ± 0.4 a	1.0 ± 0.0 a	10.5%

* In each column, the mean values followed by the same letter are not significantly different according to the Tukey test (*p* < 0.05).

## Data Availability

All new data produced in the context of this work is made available as part of this document.

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
