# Peer review of "Spontaneous Flora as Reservoir for the Survival and Spread of the Almond Anthracnose Pathogen (Colletotrichum godetiae) in Intensive Almond Orchards"

_plants, 2025, doi:10.3390/plants14121762_

Round 1
Reviewer 1 Report
Comments and Suggestions for Authors
That inoculum of Colletotrichum species can be on weeds and plants in the surroundings of orchards are already known. The present manuscript has a very long list of plant species, 15 of them were sown and investigated further. By inoculation investigations of ability to make appressoria on the different plants were evaluated. Findings of differences between the plants are interesting, but they are very little discussed in relation to differences between the plants. The importance of the results is not clearly communicated.
Specific comments:
I suggest to remove table 1 or condense it to include only the 15 species investigated further.
Table 2 is hard to read. The findings there are not much related to any characteristics of the plants. Why are there differences between the plants? And were the experiments replicated? I could not find information about that.
Figure 2, 3, 4 and 5 illustrates more or less the same. Consider combing them or remove some.
I find the conclusion a bit strange. I agree that more investigations are needed, but for every publication there should be some new information that could be highlighted.
Author Response
Comments 1: I suggest to remove table 1 or condense it to include only the 15 species investigated further.
Response 1: Table 1 is the result of compilation a list of herbaceous species (available in the flora-on database) that are present in almond orchards in the Alentejo region (Portugal). The selection of herbaceous species for this study was very difficult because of the lack of information and inventory on species present in almond orchards. Therefore, we decided to keep the compilation to allow for further work. However, we are available to place this table as supplementary material if the journal considers this pertinent in terms of the editorial style.
Comments 2: Table 2 is hard to read. The findings there are not much related to any characteristics of the plants. Why are there differences between the plants? And were the experiments replicated? I could not find information about that.
Response 2: This table condensates several values and parameters. Putting it in a less condensed form would require splitting it into several tables or graphics. Indeed, there are differences between plants regarding the way the fungus behaves on them. This is likely related to botanical differences and eventually associated biochemical or histological parameters. We have not explored these differences, but we think there is ground to further dissect these aspects. Details on the replication of the experiment were added in line 369.
Comments 3: Figure 2, 3, 4 and 5 illustrates more or less the same. Consider combing them or remove some.
Response 3: Figures 2-5 illustrate distinct aspects of spore germination, germ tube elongation, formation of appressoria and secondary conidiation. If required by the editorial style, we are happy to combine these images into a single figure containing all these images, but we think it is relevant to show all of them.
Comments 4: I find the conclusion a bit strange. I agree that more investigations are needed, but for every publication there should be some new information that could be highlighted.
Response 4: The conclusion was rewritten to highlight the main results obtained, namely the identification of species of spontaneous flora that may act as reservoirs for C. godetiae and species that limit its spread.
Reviewer 2 Report
Comments and Suggestions for Authors
The Authors investigated the potential role of weeds as a reservoire of inoculum for the pathogenic fungus Colletotrichum godetiae, the primary causal agent of almond anthracnose. They inoculated a large selection of the most frequent plant species occurring spontaneous in almond orchards in Portugal. Observations were performed by visual inspection and with scanning electron microscopy. They are clearly presented. The English style is satisfactory. The study ha epidemiological significance and can be useful for the management of anthracnose in almond orchards.
I have only minor criticisms, as it follows:
- Useabbreviated latin names of the plant and fungal species throughout the text, after the first citation of the full name.
- Line 44 (Cite formally) Colltotrichum acutatum species complex
- Line 48 Be more precise, you are specifically referring to Alentejo (Portugal), I suppose
- Line 57 Use 'warm' instead of 'high'
- Line 112 I added a reference which pertains with this study: change the numerical oder of successive references accordingly.
- Line 272 Appressoria forming conidia
- Line 273 I suppose it was three (not two): however it can be deleted
- Line 281. The expression 'pathogen evolution' seems inappropriate: may be you want to say 'the ability of the pathogen to infect the plant' or 'the fate of conidia after inoculation of plant' or something like this
- Line 284 delete 'regardless of the disinfection method'
- Line 355 Add the name of the species synonim and the respective reference (ajust the numercial order of references): the proposal of Damm et al. (Sudies in Mycology 75: 37-113) to adopt the name C. godetiae although authoritative is formally questionable as the study is successive to the article where the name C. clavatum was published and the species was separated mainly on the basis od DNA sequencing, consequently it does not correspond to the original description of C. godetiae Neerg. However as C. godetiae is the most widely accepted name a compromise could be the citation of C. clavatum as a synonim.

Author Response
Comments 1: Use abbreviated latin names of the plant and fungal species throughout the text, after the first citation of the full name.
Response 1: In this work several scientific names are used. Some may be confused if abbreviated. For instance, in line 186, if we abbreviate Trifolium vesiculosum to T. vesiculosum, this may be confused with Torilis, as the previous taxon starting with “T” (line 185) is Torilis. Because of this, we decided to use full names everywhere. Anyway, we think that in the end it should be the journal deciding how to handle this situation.
Comments 2: Line 44 (Cite formally) Colltotrichum acutatum species complex
Response 2: There are divergent views on the way to formally refer to species complex. We, and several other authors, have been following the style using non italic words without the genus name, so we would like to retain the expression “acutatum species complex”.
Comments 3: Line 48 Be more precise, you are specifically referring to Alentejo (Portugal), I suppose
Response 3: Corrected
Comments 4: Line 57 Use 'warm' instead of 'high'
Response 4: Corrected
Comments 5: Line 112 I added a reference which pertains with this study: change the numerical oder of successive references accordingly.
Response 5: Corrected
Comments 6: Line 272 Appressoria forming conidia
Response 6: Corrected
Comments 7: Line 273 I suppose it was three (not two): however it can be deleted
Response 7: Corrected
Comments 8: Line 281. The expression 'pathogen evolution' seems inappropriate: maybe you want to say 'the ability of the pathogen to infect the plant' or 'the fate of conidia after inoculation of plant' or something like this
Response 8: Corrected
Comments 9: Line 284 delete 'regardless of the disinfection method'
Response 9: Corrected
Comments 10: Line 355 Add the name of the species synonim and the respective reference (ajust the numercial order of references): the proposal of Damm et al. (Sudies in Mycology 75: 37-113) to adopt the name C. godetiae although authoritative is formally questionable as the study is successive to the article where the name C. clavatum was published and the species was separated mainly on the basis of DNA sequencing, consequently it does not correspond to the original description of C. godetiae Neerg. However as C. godetiae is the most widely accepted name a compromise could be the citation of C. clavatum as a synonim.
Response 10: We believe there is no point on perpetuating a name that is not considered in recent Colletotrichum lists of valid names, so we would like to refrain from using the name “C. clavatum” here.
Round 2
Reviewer 1 Report
Comments and Suggestions for Authors
The authors have responded to my previous comments, but not done any changes except to update the conclusion and include information about replicates. My first recommendation is continued.
Author Response
Comments 1: I suggest to remove table 1 or condense it to include only the 15 species investigated further.
Response 1: As suggested, table 1 has been condensed to show only the 15 species investigated. However, selecting herbaceous species for this study was very difficult due to the lack of information and inventory on the species present in almond orchards in Alentejo (Portugal). Therefore, we decided to make the list available as supplementary material.
Comments 2: Table 2 is hard to read. The findings there are not much related to any characteristics of the plants. Why are there differences between the plants? And were the experiments replicated? I could not find information about that.
Response 2: The results regarding germination rates and appressorium formation of conidia on 1 and 7 dai are shown in a bar chart (Figure 2) to facilitate reading and comparison of the data. Indeed, there are differences between plants regarding the way the fungus behaves on them. This is likely related to botanical differences and eventually associated biochemical or histological parameters. We have not explored these differences, but we think there is ground to further dissect these aspects. Details on the replication of the experiment were added in line 367.
Comments 3: Figure 2, 3, 4 and 5 illustrates more or less the same. Consider combing them or remove some.
Response 3: As suggested, Figures 2 and 4 were combined and Figure 3 was eliminated.
Comments 4: I find the conclusion a bit strange. I agree that more investigations are needed, but for every publication there should be some new information that could be highlighted.
Response 4: The conclusion was reworded to highlight the main results obtained, namely the identification of species of spontaneous flora that may act as reservoirs for C. godetiae and species that limit its spread.